# Expiratory Technique versus Tracheal Suction to Obtain Good-Quality Sputum from Patients with Suspected Lower Respiratory Tract Infection: A Randomized Controlled Trial

**DOI:** 10.3390/diagnostics12102504

**Published:** 2022-10-16

**Authors:** Mariana B. Cartuliares, Flemming S. Rosenvinge, Christian B. Mogensen, Thor A. Skovsted, Steen L. Andersen, Andreas K. Pedersen, Helene Skjøt-Arkil

**Affiliations:** 1Emergency Department, University Hospital of Southern Denmark, 6200 Aabenraa, Denmark; 2Department of Regional Health Research, University of Southern Denmark, 6200 Aabenraa, Denmark; 3Department of Clinical Microbiology, Odense University Hospital, 5000 Odense, Denmark; 4Research Unit of Clinical Microbiology, University of Southern Denmark, 5000 Odense, Denmark; 5Department of Biochemistry and Immunology, University Hospital of Southern Denmark, 6200 Aabenraa, Denmark; 6Department of Clinical Microbiology, University Hospital of Southern Denmark, 6200 Aabenraa, Denmark; 7Department of Clinical Research, University Hospital of Southern Denmark, 6200 Aabenraa, Denmark

**Keywords:** lower respiratory tract infection, sputum, tracheal suction, forced expiratory technique, randomized controlled trial, emergency department

## Abstract

Microbiological diagnostics of good-quality sputum samples are fundamental for infection control and targeted treatment of lower respiratory tract infections (LRTI). This study aims to compare the expiratory technique and tracheal suction on the quality of sputa from adults acutely hospitalized with suspected LRTI. We performed an open-label, randomized controlled trial. Patients were randomized to sputum sampling by tracheal suction (standard care) or the expiratory technique. The primary outcome was quality of sputum evaluated by microscopy and was analysed in the intention-to-treat population. The secondary outcomes were adverse events and patients experience. In total, 280 patients were assigned to tracheal suction (*n* = 141, 50.4%) or the expiratory technique (*n* = 139, 49.6%). Sputum samples were collected from 122 (86.5%) patients with tracheal suction and 67 (48.2%) patients with expiratory technique. Good-quality sputa were obtained more often with tracheal suction than with expiratory technique (odds ratio 1.83 [95% CI 1.05 to 3.19]; *p* = 0.035). There was no statistical difference in adverse events (IRR 1.21 [95% CI, 0.94 to 1.66]; *p* = 0.136), but patient experience was better in the expiratory technique group (*p* < 0.0001). In conclusion, tracheal suction should be considered a routine procedure in emergency departments for patients with suspected LRTI.

## 1. Introduction

Lower respiratory tract infections (LRTI) are common infectious diseases, accounting for about three million global deaths every year [1]. Targeted antibiotic treatment based on precise diagnosis is essential to avoid antimicrobial overuse and the development of antibiotic resistance. In addition, a microbiological diagnosis can have important implications for patient management and infection control measures, highlighted by the current COVID-19 pandemic. Several clinical guidelines recommend collecting a sputum sample and adjusting treatment according to identified pathogens [2,3].

Even though sputum samples provide a guide for appropriate treatment [4,5,6], the usefulness of sputum samples has been questioned, primarily due to the difficulty in obtaining good-quality sputum samples [7,8].

Sputum samples can be collected by several methods [9,10,11]. However, these methods are poorly described, and most samples are collected by self-expectoration [5]. Tracheal suction (TS) is shown to reduce contamination from the microbiota in the upper airways and is more likely to detect infectious pathogens than expectorated sputa [12,13,14] as the microbiota from the upper airways may falsely indicate a pathogen from the LRT or may overgrow the actual pathogen decreasing the diagnostic yield in culture. However, low accuracy and misclassification have also been reported [15]. In addition, patients have described the TS as painful [16], and adverse events such as hypoxia, oxygen desaturation, and mucosal bleeding have been reported [17]. The forced expiratory technique (FET) is an instructed method to facilitate expectoration that can be combined with saline inhalation to induce sputum (FETIS) [9,18,19]. Induced sputum is shown to be useful in diagnosing pulmonary tuberculosis [20]. FETIS is reported to be safe and non-invasive, but hypertonic saline and prolonged inhalation have been associated with severe adverse effects [21,22]. Previous studies have compared the different techniques in specialized departments. Generally, TS is recommended for mechanically ventilated patients to reduce the risk of infections as they have mucus retention and difficulties to cough up secretions [17]; moreover, TS can contribute to unique information on etiological agents when obtained immediately after intubation in patients with severe community-onset pneumonia [23]. FETIS has been shown to result in better prognoses in patients with a wide range of chronic respiratory diseases including cystic fibrosis, bronchiectasis, and COPD [9,10]. In the acute setting, sputum samples have important diagnostic implications as both targeted antibiotic treatment and appropriate infection control measures rely on valid microbiological results [2,3]. Poor quality samples contaminated with oropharyngeal microbiota on the other hand may lead to misleading diagnostics and inappropriate use of antibiotics. Guidelines therefore recommended only accepting good-quality samples with a low concentration of squamous epithelium from the upper airways for microbiological diagnostics [24]. This clearly underlines the importance of using the most efficient sample method.

The effectiveness of TS compared with FETIS to obtain a good-quality sputum sample has not been investigated in an emergency department (ED) setting, where the majority of patients with LRTI are seen and where safe and fast procedures, not requiring advanced skills, are requested.

This randomized controlled trial aimed to test the hypothesis that FETIS was non-inferior to (not worse than) TS in collecting good-quality sputum samples from adult patients with suspected LRTI in an acute medical ward (primary outcome). As secondary outcomes, we compared adverse events and patient experiences.

## 2. Materials and Methods

### 2.1. Study Design and Setting

This study was designed as a single-centre, non-inferiority, open-label, randomized controlled trial. The trial was conducted at Hospital Sønderjylland, which comprises two emergency departments (Aabenraa and Sønderborg) with a hospital coverage of approximately 225.000 inhabitants. A Danish ED is equivalent to an acute medical ward. The study was reported in accordance with the Consolidation Standard of Reporting Trials (CONSORT) guidelines for parallel-group randomized trials [25]. The protocol was approved by the Regional Committee on Health Research Ethics for Southern Denmark (S-20200133), registered by the Danish Data Protection Agency (20/41767) and by ClinicalTrials.gov (NCT04595526) on 20 October 2020, and completed on 5 July 2021. The statistical analysis plan (SAP) and study protocol have been published, and this publication details further information about the trial methods [26].

### 2.2. Selection of Participants

Admitted patients with suspected LRTI were consecutively identified in the patient management system (CETREA 4.2.0.0.) at the ED by a project assistant. The attending physician confirmed eligibility, and the patient’s verbal and written consent was obtained by the project assistant. Adults (>18 years of age) admitted to the ED with suspected LRTI were enrolled in the study if the attending physician identified at least one of the following pulmonary symptoms: dyspnoea, cough, expectoration, chest pain, or fever. Patients were excluded if project enrolment and sputum collection would delay urgent, lifesaving treatment (e.g., in case of severe hypoxia or cardiac events) or transfer to an intensive care unit, or if the patient had severe immunodeficiency [26].

### 2.3. Randomization and Masking

Eligible patients were randomly assigned (1:1) to either TS procedure (usual care) or FETIS (intervention). Randomization was performed by project assistants using a computer-generated randomization tool (Research Electronic Data Capture) [27], prior to collection of the sputum sample. An independent data manager generated the sequence using random block sizes of six without stratification. The data manager had no further involvement in the study. In addition, the project assistants did not have access to the randomization code, sequence, or block sizes at any time during the trial.

The study was an open-label trial as masking the intervention from participants, project assistants, or outcome assessors was impossible in the clinical setting. The statistician was blinded until data analysis was completed.

### 2.4. Interventions

Six experienced project assistants from the ED identified eligible patients; collected informed consent and patient information; and received bedside and simulation training in FET, FETIS, and TS. Furthermore, a standardized protocol for performing FETIS was developed to support consistent data collection [26]. Sputum samples were collected from patients as soon as possible or within 24 h of admission. This criterion deviated from the study protocol that stated that samples would be collected within one hour. This deviation was due to the difficulty of collecting samples in this time frame in the clinical setting. TS was performed with catheter insertion into the nares during inhalation. The catheter was gently advanced about 40 cm into the trachea, where suctioning at 200–400 mmHg was performed before withdrawing the catheter [26]. FETIS was based on the patients’ attempts to deliver a sputum sample and included FET alone and FET after sputum induction with isotonic inhalation. Efforts to minimize oropharyngeal contamination included rinsing the mouth and detailed, standard, verbal instructions in proper forced exhalation and coughing techniques [26]. Patients were instructed to deliver a sputum sample using FET. Regardless of the success of expectoration, sputum was induced using isotonic saline inhalation (0.9%) for 10 min [18], and the patient was once again instructed in FET (FETIS) [26]. Participants in the intervention group who could not deliver a sputum sample by FETIS underwent TS. These samples were not included in the intention-to-treat analysis.

### 2.5. Outcome Measures

The primary outcome was the quality of the sputum samples. The quality was defined as good or poor quality by Gram stain, and microscopy was described thoroughly in the study protocol [26]. Samples with <10 squamous epithelial cells per low power field of view (10× objective) were classified as good quality (illustrated in Figure 1), and samples with ≥10 squamous epithelial cells were classified as poor quality [28]. Sputum samples unable to be collected were considered missing as the quality could not be determined [26]. Gram stains were performed daily, and the results were generally available within 48 h.

Secondary outcomes included adverse events and patients’ experience of delivering the sputum sample. Pooled adverse events were reported and included seven variables measured before and within 10 min after sputum collection. The variables included (1) aggravation of oxygen saturation (SaO_2_) ≤93% (Chronic Obstructive Pulmonary Disease SaO_2_ ≤88%), (2) aggravation of respiratory rate (RR) ≤12/min or >20/min, (3) patient-reported aggravation of symptoms (cough, expectoration, dyspnoea, and chest pain), (4) aggravation of patient symptoms measured by Borg scale CR10, (5) occurrence of observed side effects, (6) mortality within a week, and (7) 30-day readmission. The published study protocol describes a more thorough explanation of these variables [26]. Immediately after sputum collection, the participants were requested to give a verbal score to the question: “What was your experience with this procedure?” using a five-point Likert scale ranging from “very bad”, “bad”, “neither bad nor good”, “good”, and “very good”. A visual support tool describing this scoring system was available to assist patients. Finally, participants were asked to explain the reason behind their rating [26].

### 2.6. Statistical Analysis

Statistical methods and sample size calculations are described in the SAP and study protocol for the trial. The SAP was developed and submitted before completion of recruitment, database closure, and statistical analyses [26]. To estimate the sample size, we assumed a difference between the procedures of 15% (the pre-specified margin of primary outcome). In addition, we assumed 30% missed samples in the FETIS group and 10% in the TS group. With a two-sided *p*-value, an alpha level of 5%, and a power of 84%, we would need 260 patients equally distributed between the two groups. The primary analysis followed the intention-to-treat protocol and was repeated for sensitivity purposes as a complete case analysis. The primary outcome was analysed using logistic regression. An adjusted analysis was conducted to minimize the risk of Gail’s bias [29]. Odds ratios (OR) and confidence intervals (CI) were reported. For the secondary outcomes, pooled adverse events and patient experience, we performed a Poisson regression and Wilcoxon test, respectively. Additionally, a sensitivity analysis for each type of adverse event was performed by either a chi-square test or Fischer’s exact test. Agreement of the sputum quality between FET alone and FET after sputum induction with isotonic inhalation was assessed using κ-statistics. In addition, descriptive analyses were conducted on the numbers and quality of tracheal secretions for patients in the FETIS group that could not deliver an expectorated sputum. Analyses were performed using STATA 17.0 (TX, USA). During data collection, an external assessor supervised the performance of the project assistants, and an independent microbiology expert ensured the quality of the specimen data. The project investigator monitored the daily inclusion of the patients, discussing as necessary progress with the study assistants and steering committee.

## 3. Results

In total, 534 patients were screened for eligibility between 10 November 2020 and 5 July 2021, of whom 280 (52.4%) underwent randomization. Patients were allocated to either the TS group (141 patients (50.4%)) or the FETIS group (139 patients (49.6%)) and comprised the intention-to-treat population. In the complete case analyses, 119 (85%) and 67 (48%) samples were included from the TS and FETIS groups, respectively (Figure 2).

**Figure 2 diagnostics-12-02504-f002:**
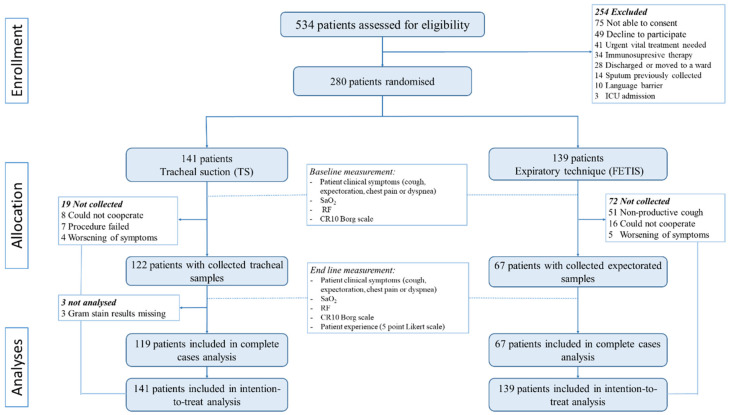
Trial profile. Randomization effectively created a balance between the two groups regarding demographic and clinical characteristics (Table 1).

The intention-to-treat analysis showed that the chance of obtaining a good-quality sputum sample was significantly higher using TS rather than FETIS (OR 1.83 [95% CI, 1.05 to 3.19]; *p* = 0.035) (Table 2). For the complete case analysis, the OR was 2.42 [95% CI, 1.31 to 4.47]; *p* = 0∙005. The sensitivity and sub-analyses are described in the Appendix A. The difference between groups was 15.06% points (58.82% for TS and 43.76% for the intervention group). There was no statistical difference when comparing the number of pooled adverse events between groups (IRR 1.21 [95% CI, 0.94 to 1.66]; *p* = 0.136) (Table 2). The sensitivity analysis showed no difference between any particular adverse event in the two groups except for bleeding (*p* = 0.0002) and dyspnoea (0.034) (Appendix A). Adverse events were reported as mild, short-lived, and without need for blood transfusions or physician consultation, except for one patient where the bleeding was reported as moderate and required physician consultation (Appendix A). There was a statistically significant difference in how patients experienced sputum collection. Patients from the FETIS group generally reported a better experience than those randomised to TS (*p* < 0.0001) (Table 2 and Appendix A). Missing data for the secondary outcome were minimal, so no imputation was necessary. The TS group had 2% and 5% missing variables for mortality and readmission, respectively, and the FETIS group had 4% missing for readmission, and both groups had 1% missing for the reported Likert scale (Appendix A). Of the 67 (48%) patients who delivered an expectorated sputum in the FETIS group, 41 (61%) produced two samples, 9 (13%) delivered a specimen only by FET, and 17 (25%) only produced a specimen by FET after sputum induction. Kappa statistics demonstrated agreement in the quality of sputum samples between FET alone (12 patients (24%)) and FET after sputum induction with isotonic inhalation (20 patients (34%)) (Kappa 0.99). A descriptive analysis of the 57 patients who could not deliver a sputum sample in the FETIS group can be seen in the Appendix A. None of the sensitivity analyses questioned the robustness of the primary results, and the variance componence yielded negligible heterogeneity.

## 4. Discussion

This study is the first randomised controlled trial comparing the quality of sputum samples collected by TS and FETIS. The result did not support our hypothesis and we showed that FETIS was inferior to TS, and TS had almost double the likelihood of ensuring a good-quality specimen. There were no differences in pooled adverse events, but FETIS was generally a more positive experience for patients than TS. The major challenge in sputum sample collection is the number of patients unable to deliver a good-quality sample [4,5,7]. This was also observed in our study, where only half of our patients delivered a sample using FETIS and less than half of these samples were of good quality despite efforts to improve expectoration.

Many studies focus on the usefulness of sputum in determining causative pathogens of LRTI but often fail to describe the sputum collection procedure adequately [5,7,30]. Different expiratory techniques positively affect secretion clearance, particularly for chronic conditions [9,10]. However, in our acute setting, FETIS did not facilitate easier secretion clearance or better-quality sputum samples. A systematic review reported that patients under instructed supervision during sputum collection delivered samples with better diagnostic value assessed by microscopy than uninstructed and unsupervised patients [31]. Therefore, our study prioritized supervision during FETIS with standardized protocols, experienced staff, and bedside training. Despite these efforts, the expiratory technique still produced sputum samples of inferior quality compared to TS.

The effectiveness of a saline solution in inducing sputum may vary with the concentration and duration of inhalation. The procedure is considered safe, with adverse events rarely reported [18]. A retrospective study focusing on patients with community-acquired pneumonia reported that a 3% saline inhalation for 30 min assisted the delivery of a quality specimen [19]. In contrast, an RCT including patients with a productive cough reported that inhalation with 3% saline for 10–15 min gave no improvement in the quality of the specimen compared to spontaneous cough [32]. The inhalation of hypertonic saline for 30–40 min was associated with dyspnoea, nausea, vomiting, and bronchoconstriction, and patients described the procedure as unpleasant, indicating a preference for bronchoscopy rather than sputum induction [33]. In our study, the low concentration of the saline solution and the duration of inhalation (≤10 min) may have contributed to the few adverse events and the low number of patients describing the procedure as unpleasant. We chose a low concentration of saline inhalation for safety and tolerance reasons [18,21], but on the other hand, a higher concentration of saline inhalation has been used in other studies resulting in more collected samples [20]; however, these studies do not compare TS and FETIS sputum sample quality. The acute setting may have played a role in the challenges of collecting sputum by FETIS such as the inclusion of patients with severe co-morbidity or unproductive coughs or, alternatively, by the short duration of inhalation or low concentration of the saline solution.

It has previously been reported that there is less contamination from the upper respiratory microbiota when specimens are collected with TS [12,13,14]. However, these studies only focus on the quality of the sputum sample and do not randomize patients or describe the collection procedure in detail. Therefore, it is difficult to determine if the quality of expectorated sputa is inferior due to the collection procedure, population, or other factors. TS is a more invasive procedure, and adverse events such as hypoxia, oxygen desaturation, arrhythmia, and mucosal bleeding have been reported for mechanically ventilated patients [17]. In addition, ICU patients report discomfort and mild pain during suctioning [16]. In our study, patients randomized to the TS group had TS-related bleedings that were assessed as minor and short-lived, confirming that pain and discomfort related to TS are the most common reasons patients report negative experiences with TS. In our study, most patients had a neutral response on the Likert scale to either FETIS or TS, and the mean difference between groups was less than one (possible type 1 error). This result may reflect that patients may be willing to undergo tracheal suction in a clinical setting despite the risk of adverse events. This study represents patients classified with mild to moderate infection according to PSI, CURB-65, and Triage. International guidelines recommend sputum collection from patients with severe LRTI [2,3], a population excluded to some degree from our study. If we extrapolate the results from this study, a routine TS procedure for frail acute patients may reduce the number of sputum sample failures.

This study aimed to investigate the efficacy of expiratory techniques regardless of the patient’s ability to expectorate. Therefore, some patients with a productive cough were allocated to the TS group, while others, unable to expectorate, were randomly assigned to the FETIS group. This random allocation is an important factor in the number of missing samples from the FETIS group. An alternative could be a multiple sample design whereby patients randomly attempt to deliver sputum samples by all three methods (FET, FETIS, and TS). However, this presents ethical challenges and may limit the generalizability of the study in an ED setting.

In contrast to other inferiority studies, we included two-sided *p*-values and CI, providing the true difference between the methods and minimizing sampling bias. There was no difference in pooled adverse events when comparing TS and FETIS, but FETIS was associated with a better patient experience but was clearly less effective in providing good-quality sputum samples supported by both the intention-to-treat and complete case analysis. Therefore, in a clinical setting, experts should not exchange TS with FETIS regardless of the benefits offered by the procedure.

The major strength of this study was the randomized controlled design, which enables us to compare the two sputum collection methods, minimizing confounding as much as possible. The variance between the staff collecting the samples was minimal with negligible heterogeneity. In addition, the standardized protocols, instructions, external supervision of the personnel, and quality monitoring of sputum samples ensured uniform data collection and increased the trial’s internal validity. These factors increase the possibility that methods and procedures are applicable to other ED contexts.

A major limitation of this study was the open-label design of the trial. However, it was not possible to blind patients, project assistants, or technicians from the two procedures. Another potential limitation of our study was the high number of patients treated with antibiotics before sampling. The diagnostic yield of sputum analysis decreases if patients have been treated with antibiotics [13,32], which has led to a debate questioning the usefulness of sputum collection. However, antibiotic treatment was evenly distributed between the TS and FETIS groups and the assessment of sputum quality based on identifying respiratory epithelial cells is not likely affected by antibiotic treatment. The decreased sensitivity of culture analysis in patients treated with antibiotics may be less of a problem when using polymerase chain reaction including multiplex, syndromic tests to diagnose LRTI, and antibiotic treatment is not likely to affect the detection of viral pathogens. However, like culture and microscopy, PCR is sensitive to contamination with upper respiratory microbiota, highlighting the importance of suitable samples. We did not measure other outcomes such as the amount of sputum or use forced expiratory volume (FEV_1_) to monitor adverse events, as recommended and assessed in other studies [21,22]. However, our study focuses on the quality of the sputum as a prerequisite to culture and further diagnostics and not airway clearance and therapy, as these studies suggest. In our setting, we did not measure FEV_1_ routinely when treating patients with acute LRTI, and the goal was a study design that reflected clinical practice.

## 5. Conclusions

Systematic reviews state that sputum samples of good quality are essential to identifying the aetiology of LRTI. In addition, clinical guidelines recommend good-quality sputum samples to support accurate LRTI diagnostics [2,3,4,5]. This study was is the first randomized controlled trial comparing the effectiveness of forced expiratory technique and tracheal suction on the quality of collected sputa in an emergency department setting. It gives useful insights into the optimal procedure to ensure the collection of good-quality sputum samples. We concluded that the forced expiratory technique is less likely to result in good-quality specimens and, therefore, is inferior to tracheal suction. In clinical practice, the implementation of TS in EDs might improve the likelihood of a correct diagnosis and the accurate treatment of LRTI. Further studies should consider multicentre locations and comparisons with other expiratory techniques and should investigate the microbiological yield from the two methods. TS should be considered a routine procedure in an ED context due to the limited value of FETIS in providing good-quality sputa, which is necessary for diagnosing LRTI.

## Figures and Tables

**Figure 1 diagnostics-12-02504-f001:**
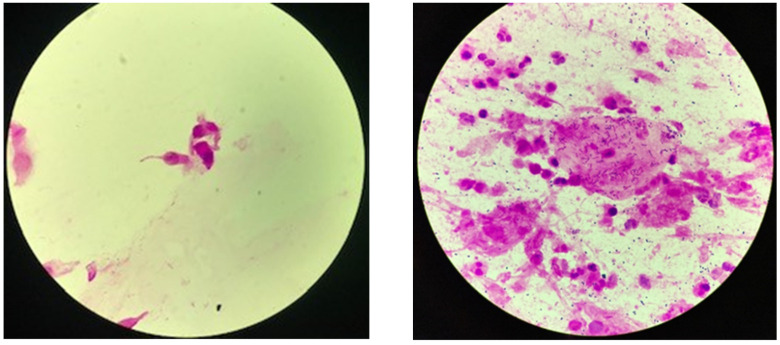
Example of Gram-stained good-quality specimens (×100 magnification). Cylindrical epithelial cell from tracheal suction (**left**) and expectorated sputum (**right**) (dense with polymorph nuclear leucocytes among Gram-positive diplococci and a single squamous epithelial cell with adhering microbiota from the upper respiratory tract). (Photo by Lomborg SA).

**Table 1 diagnostics-12-02504-t001:** Baseline characteristics of the intention-to-treat population.

	TS (*n* = 141)	FETIS (*n* = 139)	Total (*n* = 280)
Hospital Sønderjylland			
ED in Aabenraa	116 (82%)	112 (81%)	228 (81%)
ED in Sønderborg	25 (18%)	27 (19%)	52 (18%)
Sex (male)	79 (56%)	82 (59%)	161 (58%)
Age, mean years	72.9 (12.3)	71.5 (12.7)	72.2 (12.5)
Nursing home resident	8 (6%)	4 (3%)	12 (4%)
Smoking status			
Non-smokers	38 (27%)	32 (23%)	70 (25%)
Ex-smokers	76 (54%)	83 (60%)	159 (57%)
Current smokers	26 (18%)	24 (17%)	50 (18%)
Length of hospital stay ^†^, days	5.0 (2.1; 8.0)	4.0 (1.9; 6.9)	4.1 (2.0; 7.1)
SYMPTOMS			
Cough	86 (61%)	81 (58%)	167 (60%)
Expectoration	84 (60%)	77 (55%)	161 (58%)
Chest tightness	45 (32%)	49 (35%)	94 (34%)
Dyspnoea	96 (68%)	92 (66%)	188 (67%)
SEVERITY ASSESSMENT ^†^			
CURB-65 *			
Mild 0–1	62 (50%)	67 (60%)	129 (53%)
Moderate 2	43 (34%)	42 (35%)	85 (35%)
Severe 3–5	20 (16%)	11 (9%)	31 (13%)
Triage **			
Triage level 1	8 (6%)	10 (7%)	18 (7%)
Triage level 2–3	99 (71%)	105 (76%)	204 (73%)
Triage level 4–5	33 (24%)	24 (17%)	57 (20%)
Suspicion of pneumonia	99 (70%)	100 (72%)	199 (71%)
SARS-CoV-2 positive	24 (17%)	16 (12%)	40 (14%)
COMORBIDITIES ^†^			
Any comorbidity	128 (91%)	119 (86%)	247 (88%)
Respiratory diseases	66 (52%)	64 (55%)	130 (54%)
COPD ***	50 (36%)	53 (38%)	103 (37%)
Cardiovascular Diseases	85 (68%)	78 (68%)	163 (68%)
Neurological diseases	24 (19%)	25 (22%)	49 (20%)
DM ****	29 (21%)	32 (23%)	61 (22%)
Cancer	30 (21%)	23 (17%)	53 (19%)
Other diseases	60 (43%)	61 (44%)	121 (43%)
VITAL PARAMETERS			
Oxygen saturation, %	95.0 (93.0; 97.0)	96.0 (93.0; 98.0)	95.0 (93.0; 97.0)
Respiratory rate/min	21.0 (18.0; 24.0)	21.0 (18.0; 24.0)	21.0 (18.0; 24.0)
Heart rate/min	91.6 (21.6)	90.1 (17.3)	90.9 (19.6)
Systolic Blood pressure, mmHg	130.9 (20.8)	134.3 (22.6)	132.6 (21.8)
Diastolic blood pressure, mmHg	71.9 (14.5)	74.7 (16.4)	73.3 (15.5)
Fever > 38 °C	42 (30%)	45 (32%)	87 (31%)
Altered mental status	13 (10%)	9 (7%)	22 (8%)
BLOOD TESTS			
C-reactive protein, mg/L	74.0 (20.0; 168.0)	46.0 (16.0; 116.0)	54.0 (19.0; 149.0)
Leucocytes, 10^9^/L	10.8 (8.0; 14.5)	10.4 (7.5; 14.1)	10.7 (7.9; 14.2)
Neutrophilocytes, 10^9^/L	8.4 (5.9; 11.8)	7.9 (5.2; 11.0)	8.2 (5.5; 11.3)
ANTIBIOTIC TREATMENT			
Within one month	47 (33%)	48 (35%)	95 (34%)
Prior sputum collection	58 (41%)	56 (40%)	114 (41%)
Inhaled medications	33 (23%)	38 (27%)	71 (25%)

Data are *n* (%), median (IQR), or mean (SD). * CURB-65: Confusion, Urea, Respiratory rate, Blood pressure and age > 65 [26]. ** Triage: Danish Emergency Process Triage (DEPT) [26]. *** COPD: Chronic obstructive pulmonary disease. **** DM: Diabetes Mellitus I or II. ^†^ Data not available for all randomized patients.

**Table 2 diagnostics-12-02504-t002:** Results from the quality of specimens collected by FETIS and TS procedures (intention-to-treat analysis) and from adverse effects and patient experience (complete case) with FETIS as a reference group for all analyses.

**Primary Outcome**	**Unadjusted OR (95% CI)**	** *p* ** **-Value**
Quality of sputum samples	1.83 (1.05; 3.19)	0.035
**Secondary Outcome**	**Unadjusted IRR (95% CI)**	** *p* ** **-Value**
Adverse effects	1.21 (0.94; 1.66)	0.136
Patient experience of sputum collection procedure	N/A	<0.0001

## Data Availability

Due to Danish laws on personal data, data cannot be shared publicly. To request these data, please contact the corresponding author for more information. The person responsible for the research was the principal investigator and corresponding author (M.B.C.), who together with the Department of Health Research and the University Hospital of Southern Denmark, owns the data and has access to the final data set.

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
