# Peer review of "Expiratory Technique versus Tracheal Suction to Obtain Good-Quality Sputum from Patients with Suspected Lower Respiratory Tract Infection: A Randomized Controlled Trial"

_diagnostics, 2022, doi:10.3390/diagnostics12102504_

Round 1
Reviewer 1 Report
Dear authors,
This manuscript compares two techniques for obtaining sputum samples. It is interesting to know whether a less invasive technique yields promising results.
It is an interesting proposal although I have a number of questions to resolve:
INTRODUCTION
- The introduction is very short. It would be interesting to have additional information to support the authors' hypotheses. Authors should include some epidemiological data to support their claims.
- The target should be corrected. Authors should clearly state whether their objective was based on which of the two techniques they consider to be better.
MATERIALS AND METHODS
2.2. Selection of Participants:
- What was the target population? How was the sample chosen? The authors must specify it.
- Line 73: I don´t understand the exclusion criterion: "participation delayed urgent treatment". The authors should explain this topic better.
DISCUSSION
- The authors have not explained how to apply the research to clinical practice.
- The last paragraph should be placed as a "Conclusions" section.
REFERENCES
- Many bibliographies are obsolete. The bibliographic citations used are more than 5 years old (74.1 %). The authors must update and arrange the bibliography.
Reviewer 2 Report
This article estimates the expiratory technique versus tracheal suction for a high-quality sputum from patients with suspected low respiratory infection. The authors should obtain more information regarding the amount of sputum and the bacterial pathogens identified in culture according to differential quality criteria and 3mo mortality rate. Presence of side effects from the procedure as arterial pressure and cardiac arrythmias should be add in the text. In addition, neb antibiotics or FEV1 values mainly for copd or asthmatic patients can be mentioned.
Reviewer 3 Report
The authors aim to investigate for the first time the differences between the two techniques for taking samples of respiratory sputum samples for LRTI. They performed an open-label randomized clinical trial. Statistics are appropriate and the methodology is well described in a prior publication. They measured the safety outcomes and quality of the sputum.
Concerns on this paper:
The authors mention on line #38 the COVID-19 pandemic and its importance, also on Table 1 mentioned SARS-Co2 Testing up to 14% of samples. My question is how the methodology was adjusted during the COVID-19 pandemic and how they handled asymptomatic patients with COVID enrolled in the study.
Did the authors collect data on the microbiology /viral profiling recovered between both techniques? Will be interesting to have this data on a table.
The authors used saline 0.9% for induction of sputum. Previous studies used 3% or 3.5%. Could you explain why not use hypertonic saline?
The abstract could be improved: Specifically on the results and conclusion sentences.
Minor typos should be corrected. Ex. Cardiovaskular on table 1.
